# The Pathogenetic Role of RANK/RANKL/OPG Signaling in Osteoarthritis and Related Targeted Therapies

**DOI:** 10.3390/biomedicines12102292

**Published:** 2024-10-10

**Authors:** Gabriele Di Cicco, Emanuela Marzano, Andrea Mastrostefano, Dario Pitocco, Rodrigo Simões Castilho, Roberto Zambelli, Antonio Mascio, Tommaso Greco, Virginia Cinelli, Chiara Comisi, Giulio Maccauro, Carlo Perisano

**Affiliations:** 1Department of Physiology and Pharmacology, Sapienza University of Rome, 00185 Rome, Italy; gbrldicicco@gmail.com (G.D.C.);; 2Diabetes Care Unit, Endocrinology, University Hospital “A. Gemelli”, Catholic University of the Sacred Heart, 00136 Rome, Italy; 3Department of Orthopaedics and Traumatology, Mater Dei Hospital, Belo Horizonte 30170-041, Brazil; 4Department of Orthopedics and Geriatric Sciences, Catholic University of the Sacred Heart, 00136 Rome, Italy; 5Department of Orthopedics and Rheumatological Sciences, Fondazione Policlinico Universitario A. Gemelli IRCCS, 00136 Rome, Italy; 6Department of Life Sciences, Health, and Healthcare Professions, Link Campus University, 00165 Rome, Italy

**Keywords:** osteoarthritis, joint disease, RANKL, OPG, targeted therapies, hyaluronic acid, denosumab

## Abstract

**Background:** Osteoarthritis (OA) is the most common degenerative joint disease and affects millions of people worldwide, particularly the elderly population. The pathophysiology of OA is complex and involves multiple factors. **Methods:** Several studies have emphasized the crucial role of inflammation in this process. The receptor activator of NF-κB ligand (RANKL), the receptor activator of NF-κB (RANK), and osteoprotegerin (OPG) trigger a signaling cascade that leads to the excessive production of RANKL in the serum. **Conclusions:** The aim of this narrative review is (i) to assess the role of the RANK/RANKL/OPG signaling pathway in the context of OA progression, focusing especially on the physiopathology and on all the mechanisms leading to the activation of the inflammatory cascade, and (ii) to evaluate all the potential therapeutic strategies currently available that restore balance to bone formation and resorption, reducing structural abnormalities and relieving pain in patients with OA.

## 1. Introduction

Osteoarthritis (OA) is the most common degenerative joint disease and affects millions of people worldwide, particularly the elderly population [1,2]. OA has multifactorial causes such as genetic, environmental, and mechanical factors. Indeed, advanced age, obesity, surgical joint treatments, repeated joint injuries, genetic predisposition, and metabolic disorders have been detected as risk factors for OA development [3,4]. Moreover, it has been shown that aging is a significant factor in the gradual deterioration of joint function. Pathological changes include a decline in articular cartilage, synovial inflammation, the advancement of degradation, subchondral bone (SB) sclerosis, and the formation of bone spurs [5].

OA has several symptoms that lead to medium- and long-term physical disability. About 30% of the population of the United States experiences joint pain and functional limitations [6]. Its social, personal, and economic impacts cause high public health expenditure. It is estimated that about 240 million people worldwide have symptomatic OA, with global numbers increasing since 1990 by 9.3% for hip OA and 8.2% for knee OA. According to some authors, the global incidence of the disease is estimated to be about 20% of the population [7], but recent epidemiological studies have shown many regional and individual differences between countries. For example, in Europe, the incidence is about 10–17%; in North America, it is 12–21%; and in Asia and Africa, it is 16–29% [8,9]. Considering the statistical differences, OA is influenced by different genetic and environmental risk factors, and this may contribute to the specificity of epidemiological studies (VII) [10]. Due to its non-specific symptoms, there are still many problems relating to its definition. For example, Kellgren and Lawrence (KL) originally described a score based on antero-posterior (AP) radiographs of the knee using values from 0 to 4, where 0 indicates no OA and 4 indicates severe OA. This classification also provides detailed descriptions of the pathology, such as osteophyte formation, reductions in articular cartilage associated with sclerosis of the SB, and the formation of pseudo-cystic areas in the SB. According to the WHO, a Lawrence, Bremner, and Bier radiological osteoarthritis score of 2–4 is still the most widely used radiological classification in epidemiological studies [11,12].

In the 1990s, the American College of Rheumatology developed a classification of OA that distinguished several features, including clinical, laboratory, and/or radiologic features. The goal of this classification was to differentiate patients with OA from those with other joint diseases. Although the criteria were non-specific, the purpose of the study was to separate symptomatic arthritis from asymptomatic arthritis evidenced solely by radiographic changes. Therefore, Altman et al. planned to develop criteria for single joints. Specifically, for the knee, they used combinations of clinical data, clinical plus laboratory data, and clinical plus laboratory plus radiographic data. Meanwhile, for the hand, they used clinical data only and a combination of clinical data plus radiographic data. Following the same line of thinking, for the hip, they used clinical data with or without laboratory data and a combination of clinical, laboratory, and radiographic data [13,14,15].

The predominant clinical sign is pain. The clinical manifestation can be discontinuous, and it typically worsens with weight bearing and physical activity. The patient typically reports morning stiffness; this condition is related to prolonged inactivity but it resolves in less than 30 min, unlike in rheumatoid arthritis, in which this symptom persists over time [16]. In addition, reduced movement may be associated with disturbed moods and depression, difficulty sleeping, and more generally decreased quality of life [17]. Correlations between local, systemic, immunochemical, and phenotypic factors are required to identify the treatment target.

The treatments used to slow down OA progression are focused only on pain reduction, ignoring the main cause. Today, efforts are being made to introduce new solutions that aim to reduce OA progression. Recent developments have emphasized the importance of the nuclear factor-kappa B/receptor regulator of the nuclear factor-kappa B ligand/osteoprotegerin (RANK/RANKL/OPG) signaling pathway in the onset and advancement of OA [18].

The pathogenetic role of this signaling pathway is due to its regulation of bone remodeling, which affects both chondrocytes and osteoclasts. Additionally, the RANK/RANKL/OPG pathway is involved in osteoclast differentiation from osteoblasts, mediated by inflammatory cytokines, growth factors, and mechanical loading [19,20].

Clinical studies carried out in patients with OA have demonstrated an elevated bone resorption index in the early stage of the disease. These findings are supported by in vivo findings of studies performed on OA animal models that allowed for a chronological analysis of disease progression. In particular, an increased number of osteoclasts, along with surface and trabecular thickness reductions in the SB, was observed in the early stage of the disease. In contrast, active bone formation only occurs several months after OA induction, typically during the advanced stage [21,22].

Moreover, other clinical studies classified patients affected by OA into two groups identified by the levels [low (L) or high (H)] of endogenous prostaglandin E2 (PGE2) in their osteoblasts. The L-OA and H-OA groups were identified by the presence of two distinct phenotypes of SB osteoblasts. Assessments of SB thickness revealed a reduction in L-OA patients and an increase in the case of H-OA, as compared to unaffected people. Furthermore, abnormal expression levels of the bone remodeling factor OPG and RANKL were revealed in both L-OA and H-OA osteoblasts compared to unaffected ones, suggesting an increase in SB resorption in the L-OA as a consequence of the OPG/RANKL ratio reduction, whereas a shift toward bone formation was observed in H-OA due to an increased OPG/RANKL ratio.

A deeper comprehension of the pivotal role played by the RANK/RANKL/OPG pathway in OA onset is essential. Recent research has highlighted the complexity of the molecular, cellular, and environmental factors contributing to OA pathogenesis, leading to the proposal of innovative therapies such as monoclonal antibodies and hyaluronic acid to target RANK/RANKL/OPG signaling [23].

This narrative review aims to describe the biochemical signal transduction underlying OA onset and to provide a comprehensive overview of clinical innovations targeting the RANK/RANKL/OPG signaling pathway.

## 2. RANK/RANKL/OPG Signaling in Osteoarthritis

As a key regulator of multiple important aspects of OA, including cartilage degradation, aberrant SB remodeling, and synovium inflammation, the RANK/RANKL/OPG signaling pathway plays a significant role in the pathogenesis and progression of osteoarthritis [24,25]. Dysfunction within this pathway results in an imbalance between bone formation and resorption, leading to osteolytic lesions and structural abnormalities in the SB, which contribute to subchondral sclerosis and are accompanied by localized inflammation [26,27]. However, exogenous RANKL released by the chondrocytes may not be directly linked to cartilage breakdown or chondrocyte activation, nor does it activate nuclear factor kappa-B (NF-κB) or the transcription of proinflammatory cytokine genes. Osteoblasts exhibiting higher metabolic activity and lower RANKL levels become more prevalent in the development of OA [28]. The RANKL/RANK/OPG signaling pathway is mainly involved in the intricate molecular mechanism that underlies aberrant SB remodeling, especially in pathological states such as OA. The overexpression of RANKL is a crucial starting point for OA. In osteoclast precursor cells, the excess RANKL interacts with its corresponding receptor, RANK. The differentiation of these precursors into fully mature osteoclasts is triggered by a complex cascade of events. Mature osteoclasts, which have a strong capacity to resorb bone, are the cause of the accelerated bone resorption rate in the SB compartment. This imbalance between bone formation and resorption is responsible for the characteristic structural anomalies found in the subchondral region [29,30]. Abnormal SB remodeling can stimulate osteoclasts to release substances such as Netrin-1, a molecule critical for promoting sensory nerve axonal growth within the SB environment. Sensory nerve fibers within the SB are involved in the modulation of pain sensitivity, which is a prominent feature of OA. These fibers, which are rich in nociceptors, detect and transmit pain signals from the SB to the central nervous system [31]. This mechanism significantly influences both structural changes and pain perception, which is a critical consideration in the clinical management of OA.

The distinct features of osteoarthritis (OA), including the loss of trabecular bone, the presence of cyst-like lesions, and the occurrence of sclerotic changes, offer clear indications of these abnormalities. However, the complex molecular framework is not confined to the enhancement of osteoclastogenesis and bone resorption. The bidirectional attack on osseous tissue further exacerbates the remodeling abnormalities of the SB. Elevated levels of Tumor Necrosis Factor-α (TNF-α), Interleukin-1α (IL-1α), Interleukin-6 (IL-6), and pro-inflammatory cytokines, commonly found in OA, induce osteoclastogenesis directly by targeting specific receptors of osteoclasts or indirectly through the RANK/RANKL/OPG signaling pathway [32,33,34,35]. Consequently, inflammatory factors and RANKL work together to increase osteoclast activity and bone resorption, thereby worsening the structural abnormalities associated with OA. Conversely, anti-osteoclastogenic cytokines, such as interferon-gamma (IFN-γ), significantly inhibit RANK signaling, thus affecting osteoclastogenesis [36,37]. Bone remodeling in the SB is regulated by the RANK/RANKL/OPG signaling pathway, which also involves β-catenin release by chondrocytes through the Wnt/β-catenin system. This system regulates the function of subchondral osteoclasts via osteochondral crosstalk channels, contributing to bone reorganization. Moreover, Wnt/β-catenin signaling regulates OPG expression in osteoblasts [38,39]. A decrease in bone mass and volume can be directly linked to the overexpression of RUNX2, a key regulator of bone formation, which is associated with the depletion of β-catenin due to its inhibition of the canonical Wnt/β-catenin signaling pathway. The suppression of RUNX2 leads to increased RANKL expression and reduced OPG expression. In fact, activating β-catenin can reverse the high bone resorption in the SB caused by RUNX2 overexpression in mice, a process closely linked to the RANKL/OPG signaling pathways [27] (Figure 1).

The integrity of articular cartilage is indirectly influenced by chronic inflammation and bone resorption, both of which are regulated by the RANK/RANKL/OPG signaling pathway. The elevated expression of RANKL within the cartilaginous matrix promotes a cascade of molecular events that contribute to the progressive deterioration of the OA joint. RANKL interacts with its receptor, RANK, which is expressed on the surface of osteoclast precursor cells. The activation of the NF-κB pathway, triggered by the interaction between RANKL and RANK, leads to the formation of specialized osteoclasts responsible for bone resorption. In OA, heightened levels of RANKL result in a marked increase in the number and activity of osteoclasts within the SB [40,41,42]. The morphological changes in the SB, due to osteoclast hyperactivity, cause an abnormal biomechanical environment in the joint, which further exacerbates damage to the articular cartilage [43]. When OA affects cartilage, RANKL upregulation can lead to abnormal signal transduction pathways in the chondrocytes, the primary cellular unit of articular cartilage. Signaling disruption promotes the abnormal production of extracellular matrix components, including collagen and proteoglycans, resulting in the loss of structural integrity in cartilage. This contributes to the progressive degradation of the cartilage [44]. The signaling pathways activated by RANKL in chondrocytes are directly associated with apoptosis, leading to a decrease in the chondrocyte population and a decline in their capacity to maintain cartilage homeostasis [45]. In OA patients, the synovial tissue exhibits a significant increase in RANKL expression, as compared to that in healthy individuals. This upregulation triggers inflammation in the affected joints, marking the initiation of the inflammatory process [46]. The onset of the inflammatory response plays a crucial role in the RANKL signaling pathway, prolonging inflammation in the synovial environment. In the progression of OA, the overexpression of RANKL stimulates osteoclast activation within the joint microenvironment [47]. This multifaceted process results in the degradation of articular bone, driven by bone resorption and amplified by the inflammatory milieu, which also releases bioactive factors that further intensify the inflammation [48]. The bidirectional relationship between RANKL-induced osteoclast activation and synovial inflammation leads to pathological changes and accelerates the degenerative processes associated with OA. Moreover, the production of pro-inflammatory cytokines is intricately linked to RANKL signaling, which, together with RANKL, creates a pro-inflammatory environment around the synovium and extends its effect to surrounding structures. RANKL and these proinflammatory cytokines play a pivotal role in the progression of OA, promoting cartilage degradation, synovial hypertrophy, and angiogenesis, all of which are hallmarks of this disease [49] (Figure 2).

## 3. Therapeutical Implications on the RANKL/RANK/OPG Signaling Pathway

OA leads to a progressive decline in bone density and structural integrity in the affected joints. This deterioration is primarily driven by the activation of RANKL, which facilitates the differentiation and activity of osteoclasts [50]. Beyond compromising the mechanical stability of the joint, this process of bone resorption also releases inflammatory mediators that were previously sequestered within the bone matrix, thereby exacerbating synovial inflammation. Furthermore, this process generates products that may act as damage-associated molecular patterns (DAMPs), triggering heightened immune responses and inflammation [51].

Innovative therapeutic strategies have emerged from the understanding of the critical role played by the RANKL/RANK/OPG signaling pathway in orchestrating this complex pathophysiological mechanism. Modulation of this pathway is currently being explored to improve bone remodeling abnormalities and reduce the pain burden in OA patients. To restore balance in the bone resorption process, therapies involving anti-TNFα antibodies and osteoclast inhibitors are being utilized. These treatments aim to slow bone loss and, consequently, mitigate structural anomalies, potentially alleviating the pain experienced by individuals affected with OA by targeting the RANKL/RANK/OPG signaling pathway [29,30].

### 3.1. Hyaluronic Acid

Hyaluronic acid (HA) was first isolated in 1934 from the vitreous body of bovine species. HA is a naturally occurring glycosaminoglycan with high viscosity, consisting of unbranched chains of repeating disaccharide units composed of D-glucuronic acid and N-acetyl-D-glucosamine [52,53]. It is one of the main components of the extracellular matrix (ECM) [54]. While most of the body’s HA is found in the skin, it plays a crucial role in the joints, where it functions as a lubricant, stabilizer, shock absorber, and osmoregulatory agent [55].

In osteoarthritis (OA), inflammation leads to the depolymerization of HA and alterations to its molecular weight, concentration, and cross-linking [56]. Intra-articular supplementation of exogenous HA, known as viscosupplementation, is effective in the management of OA, primarily aiming to alleviate pain [57,58].

The various preparations for viscosupplementation differ in their molecular weight, viscosity, and concentration. Exogenous HA plays a crucial role in blocking the pathological mechanisms of osteoarthritis, stimulating chondrocyte synthesis, and promoting cartilage regeneration. Additionally, HA decreases the formation of pro-inflammatory agents and matrix metalloproteinases (MMPs), both of which contribute to cartilage breakdown and inflammation [59]. HA can also be combined with platelet-rich plasma (PRP) to exert a synergistic effect in inhibiting the inflammatory cascade, reducing the activity of NF-κB [59,60,61]. Another potential combination is HA with polynucleotides (PNs), which enhance collagen production and tissue regeneration and reduce inflammation, thereby offering additional functional benefits [62].

Nakao et al. demonstrated that HA in bone marrow cells supports RANKL-mediated osteoclast activity through the regulation of the calcitriol receptor and signal transducer and activator of transcription 3 (STAT3) signaling pathways [63].

### 3.2. Denosumab

Denosumab is a human monoclonal antibody of the IgG2 class, engineered from Chinese hamster ovary (CHO) cells [64,65].

Its primary target is RANKL, a cytokine that, by interacting with the RANK receptor on the membrane of pre-osteoclasts and mature osteoclasts, promotes their formation, activation, and survival. Denosumab inhibits osteoclast-mediated bone resorption, thus increasing bone strength [66,67].

The pharmacokinetic properties of denosumab are notable. Following a 60 mg subcutaneous injection administered biannually, a peak serum concentration (Cmax) of 6 µg/mL is typically reached approximately 10 days post-injection. After the Cmax is achieved, serum concentrations decline with a half-life of approximately 26 days over the subsequent three months. In the majority of patients, denosumab serum concentrations become undetectable after six months. The degradation pathway of denosumab remains unknown and is a topic of ongoing debate. The drug’s metabolism and elimination are likely related to immunoglobulin clearance. The pharmacokinetics of denosumab are not affected by the formation of anti-drug antibodies, nor are they influenced by gender or disease status. While body weight may have a moderate impact, it is not clinically significant [68].

Additionally, denosumab’s pharmacokinetics are not expected to be altered by hepatic impairment, nor does it interfere with the cytochrome P450 enzyme system, including CYP3A4 [68,69].

Denosumab has been approved by both the FDA and EMA for several conditions, including osteoporosis in postmenopausal women, bone metastases, and prostate and breast cancer.

Recently, denosumab, along with other drugs used in osteoporosis treatment, has been explored for its potential in osteoarthritis therapy, yielding promising results. It has shown efficacy in reducing joint erosion [70]. A 48-week randomized controlled trial demonstrated that denosumab effectively slows the progression of hand joint erosion [69].

In conclusion, an editorial paper highlights the superiority of pharmacological therapies that target the inflammatory cascade triggered by the immune system over regenerative therapies in the treatment of osteoarthritis [71].

### 3.3. Antioxidants

In vitro studies have demonstrated that antioxidants have the capacity to protect osteoclasts and osteoblasts from oxidative stress; Moreover, reducing oxidative stress can prevent chondrocyte damage and cartilage degeneration [72]. A notable example is N-acetylcysteine (NAC), which has been shown to protect chondrocytes from oxidative stress induced by Interleukin-1 (IL-1), a cytokine that also plays a significant role in promoting chondrocyte apoptosis [20].

### 3.4. Bisphosphonates

Bisphosphonates are a class of pharmacological agents designed to inhibit bone resorption. They achieve this by binding to hydroxyapatite crystals within the bone, which leads to the apoptosis of osteoclasts, the cells responsible for bone degradation. By reducing the osteoclasts’ activity and lifespan, bisphosphonates effectively decrease bone resorption and turnover, thus helping to maintain bone density [73,74].

Bisphosphonates do not directly target the RANK/RANKL/OPG pathway. Instead, they primarily exert their effects through the inhibition of osteoclast activity and induction of osteoclast apoptosis. However, by reducing osteoclast activity, bisphosphonates indirectly affect the dynamics of RANKL and OPG levels [75]. For example, the reduction in bone resorption associated with bisphosphonate therapy can lead to changes in the local concentrations of RANKL and OPG, although this effect is less direct compared to that of drugs specifically targeting the RANK/RANKL interaction [76].

### 3.5. Anti-Inflammatory Agents

Anti-inflammatory agents, including nonsteroidal anti-inflammatory drugs (NSAIDs) and corticosteroids, are commonly used to manage inflammation and pain in various musculoskeletal disorders, particularly osteoarthritis [77].

NSAIDs exert their effects by inhibiting cyclooxygenase enzymes (COX-1 and COX-2), which are crucial for the synthesis of prostaglandins—lipid compounds that mediate inflammation, pain, and fever. By reducing prostaglandin levels, NSAIDs alleviate inflammation and pain [78]. By reducing inflammation, NSAIDs can influence the local environment of bone and cartilage. Chronic inflammation is known to upregulate RANKL expression and increase osteoclast activity. Therefore, by controlling inflammation, NSAIDs may indirectly affect the RANK/RANKL/OPG pathway and reduce osteoclast-mediated bone resorption. However, the direct impact of NSAIDs on RANK/RANKL/OPG signaling remains inadequately established.

In contrast, corticosteroids mimic the effects of hormones produced by the adrenal glands and have potent anti-inflammatory properties. They inhibit multiple inflammatory pathways by suppressing the production of pro-inflammatory cytokines and other mediators [79]. These drugs can have a more direct effect on bone metabolism. They can alter the balance between RANKL and OPG by increasing RANKL expression and decreasing OPG production, thereby promoting osteoclastogenesis and bone resorption. While effective in reducing inflammation, the prolonged use of corticosteroids can lead to bone loss and osteoporosis, partly due to their impact on the RANK/RANKL/OPG pathway.

## 4. Discussion

OA is a multifactorial degenerative disease affecting the entire joint, characterized by complex pathogenesis. Multiple factors contribute to the onset of OA, including excessive mechanical loading, inflammatory mediators, aging, and other elements. These factors lead to distinct early-stage OA manifestations with specific molecular signaling signatures. The degradation and erosion of articular cartilage, excessive growth and inflammation of synovial tissue, abnormal blood vessel formation in the synovial joint, disruption of subchondral bone, and instability of ligaments and tendons may collectively or individually contribute to the development and progression of this condition.

In this review, we comprehensively explored the RANK/RANKL/OPG signaling pathway and its pivotal role in the regulation of bone remodeling and the onset of osteoarthritic processes. Our narrative review provides an in-depth analysis of the key pathophysiological and cellular mechanisms associated with this pathway, drawing from a thorough examination of the existing literature.

The strength of our study lies in its detailed synthesis of the cellular mechanisms, cytokine cascades, and current pharmacological treatments related to the RANK/RANKL/OPG pathway. By integrating these aspects, we offer a holistic view of the pathway’s involvement in both normal bone homeostasis and pathological conditions such as osteoarthritis. Our review highlights the intricate interplay between these signaling molecules and their impact on bone health and disease progression.

Several studies have emphasized the crucial role of the RANK/RANKL/OPG signaling pathway in regulating the onset and progression of OA. By either increasing RANKL production or decreasing the levels of its decoy receptor OPG, an imbalance is created, favoring osteoclast activation and bone resorption [80].

OPG’s dual roles as a decoy receptor and a lifespan regulator account for its suppression of RANKL. Maintaining an appropriate balance between RANKL and OPG is critical for potential therapeutic applications involving OPG. Achieving the most effective therapeutic approach would benefit from maintaining an optimal concentration of soluble OPG while minimizing its internalization process [23].

Moreover, it has been proven that RANKL’s influence extends beyond bone regulation. It is crucial in regulating immune response and has a significant influence on the inflammatory environment. Aberrant RANKL signaling stimulates the production of inflammatory cytokines, which are recognized as key drivers of inflammation in OA [75]. Denosumab, which acts on the RANK/RANKL/OPG pathways, represents a promising prophylactic treatment for early-stage OA, as it can arrest inflammatory processes and slow the progression of the disease [69,70,71].

Hyaluronic acid (HA) is classified into four molecular weight (MW) ranges: low MW (<500 kDa), intermediate MW (600–800 kDa), high MW (800–3000 kDa), and ultra-high MW (>3000 kDa). Low-to-intermediate-MW HA is particularly susceptible to degradation by the free radicals, enzymes, and cytokines found in the synovial fluid of arthritic joints, although it offers higher nutritional value [81,82]. In contrast, increases in MW improve the stability and extension of the HA molecular chain, making it more similar to the natural HA found in healthy joints, where the MW typically ranges from 4000 kDa to 5000 kDa. Clementi et al. demonstrated that ultra-high-MW HA is safe and effective for treating specific types of osteoarthritis, showing greater efficacy in reducing pain and disability compared to intermediate-MW HA [83,84].

Researchers are also exploring innovative treatments such as gene therapy, which utilizes adenovirus or lentivirus techniques to regulate the RANK/RANKL/OPG pathways. Other approaches involve the use of liposomes, nanoparticles, and electroporation as vectors to deliver genes affecting joint health [84,85].

OA represents a significant social, medical, and economic burden; in fact, the incidence of this disease is leading to a year-by-year increase in global health care spending. Several studies have also shown that patients with OA have a higher risk of hospitalization and a higher rate of emergency department access [86,87,88].

However, it is important to acknowledge that certain mechanisms within this pathway remain poorly understood. Despite our extensive review, some aspects of the RANK/RANKL/OPG signaling cascade and its role in osteoarthritic processes continue to be ambiguous in the literature. This underscores the need for further research to elucidate these unclear mechanisms and to refine our understanding of how they contribute to bone remodeling and osteoarthritis.

## 5. Conclusions

This review thoroughly explored the RANK/RANKL/OPG signaling pathway in osteoarthritis (OA) and its modulation for targeted therapies. Undoubtedly, further targeted strategies are essential to prevent or mitigate OA, a condition that presents significant challenges. These advancements offer promising prospects for enhancing the quality of life of OA patients globally.

## Figures and Tables

**Figure 1 biomedicines-12-02292-f001:**
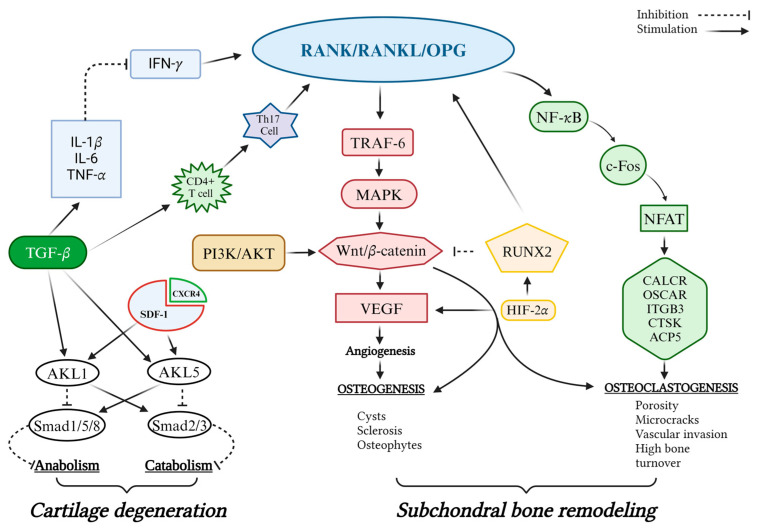
Potential mechanisms underlying signaling pathway crosstalk in OA. The RANK/RANKL/OPG pathway regulates the balance between bone formation and resorption, maintaining bone density and strength.

**Figure 2 biomedicines-12-02292-f002:**
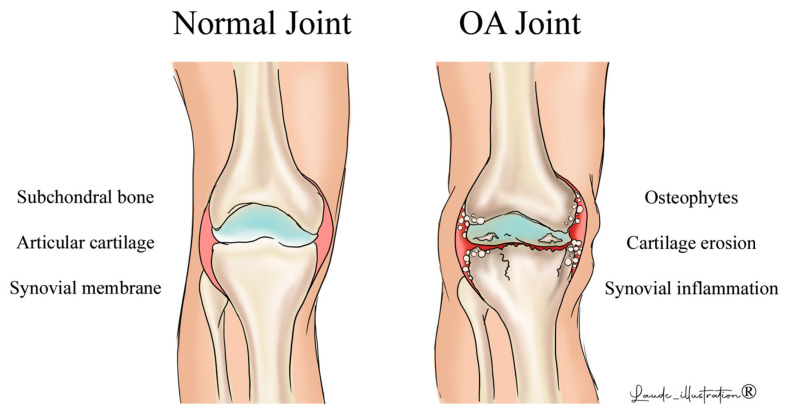
Clinical evidence of different phenotypes of OA. The image shows the difference between a healthy joint and one affected by osteoarthritis. The main pathophysiological processes that occur in osteoarthritis are illustrated in the figure.

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
