# Peer review of "The Pathogenetic Role of RANK/RANKL/OPG Signaling in Osteoarthritis and Related Targeted Therapies"

_biomedicines, 2024, doi:10.3390/biomedicines12102292_

Round 1

Reviewer 1 Report

Comments and Suggestions for Authors

The authors assessed the role of RANK/RANKL/OPG signalling pathway in the contest of the osteoarthritis (OA) progression and evaluated all potential therapeutic strategies, which is an interesting summary of the pathogenesis and treatment options for OA. The conclusion showed that the RANK/RANKL/OPG signalling pathway plays an important role in the targeted treatment of osteoarthritis and have a certain reference role in the treatment of osteoarthritis. Overall, this review provides foundational insights into the mechanism of OA, but major revisions are needed before publication can be considered.

1.        For the convenience of readers, it is suggested that the author should strengthen his writing ability to improve the quality of manuscripts.

2.        In line 46, abbreviations need to be written out in full only on the first occurrence, and thereafter abbreviations are used instead.

3.    In the Introduction section, the author did not mention the treatment options for osteoarthritis, so it is hoped that the author can briefly describe the traditional and current treatment options for osteoarthritis.

4.    In the Figure 1, 2 and 3 sections, In order to enable readers to better understand the information in the pictures, please use brief words to describe the content of the pictures.

5.    In the Figure 2 section, the text in the picture is too small. In order to give readers a better reading experience, It is suggested that the author can make appropriate adjustments to the picture.

6.    In Discussion section, the author should provide a full discussion of the review. It is recommended that the authors discuss both limitations and the role of this study in other fields.

Author Response

Full comment : The authors assessed the role of RANK/RANKL/OPG signalling pathway in the contest of the osteoarthritis (OA) progression and evaluated all potential therapeutic strategies, which is an interesting summary of the pathogenesis and treatment options for OA. The conclusion showed that the RANK/RANKL/OPG signalling pathway plays an important role in the targeted treatment of osteoarthritis and have a certain reference role in the treatment of osteoarthritis. Overall, this review provides foundational insights into the mechanism of OA, but major revisions are needed before publication can be considered.

Response: Thank you for pointing this out. We have carefully read your comment, and we agree with your suggestions. Therefore, we have tried to make the appropriate changes. 

Comment 1: For the convenience of readers, it is suggested that the author should strengthen his writing ability to improve the quality of manuscripts.

Response 1: We agree with your opinion. We have tried to improve the text's quality. 

Comment 2: In line 46, abbreviations need to be written out in full only on the first occurrence, and thereafter abbreviations are used instead.

Response 2: Done. Please, see line 86-87

Comment 3: In the Introduction section, the author did not mention the treatment options for osteoarthritis, so it is hoped that the author can briefly describe the traditional and current treatment options for osteoarthritis.

Response 3: Thank you for your observations. We have expanded the section on treatment in the text to provide the reader with a broader view of everything we found in the literature on the topic.

Comment 4: In the Figure 1, 2 and 3 sections, In order to enable readers to better understand the information in the pictures, please use brief words to describe the content of the pictures.

Response 4: We have added a brief caption describing the figures.

Comment 5: In the Figure 2 section, the text in the picture is too small. In order to give readers a better reading experience, It is suggested that the author can make appropriate adjustments to the picture.

Response 5:  We have modified the images. We hope they are now clearer and easier to view and read.

Comment 6: In Discussion section, the author should provide a full discussion of the review. It is recommended that the authors discuss both limitations and the role of this study in other fields.

Response 6: We have revised the discussion to make it more comprehensive. We have highlighted the changes/additions made in the text.

Reviewer 2 Report

Comments and Suggestions for Authors

Thank you for the opportunity to review the manuscript entitled "The Pathogenetic Role of RANK/RANKL/OPG Signalling in Osteoarthritis and Related Targeted Therapies." This review focuses on osteoarthritis (OA), the most common degenerative joint disease affecting millions worldwide, especially the elderly. It highlights the multifactorial pathogenesis of OA, emphasizing the role of inflammation and the RANKL-RANK-OPG signaling pathway in the early acute phase and evaluates potential therapeutic strategies to balance bone formation and resorption, reduce structural abnormalities, and alleviate pain in OA patients.

I recommend the following statements for improvement:

  1. Please reduce the similarity, as indicated in the attached document.
  2. Clarify whether the included figures are created by the authors or sourced from other papers. If the latter, please state whether you have permission to use them or cite the original sources.
  3. For therapeutic treatments with hyaluronic acid and denosumab, I suggest including a diagram to enhance understanding of their roles in the therapeutic implications of OA.
  4. Please include at least two more therapeutic treatments or drugs, or biomaterials for a more exhaustive review.

After these major changes, a further review will be necessary.

Author Response

Full Comment: Thank you for the opportunity to review the manuscript entitled "The Pathogenetic Role of RANK/RANKL/OPG Signalling in Osteoarthritis and Related Targeted Therapies." This review focuses on osteoarthritis (OA), the most common degenerative joint disease affecting millions worldwide, especially the elderly. It highlights the multifactorial pathogenesis of OA, emphasizing the role of inflammation and the RANKL-RANK-OPG signaling pathway in the early acute phase and evaluates potential therapeutic strategies to balance bone formation and resorption, reduce structural abnormalities, and alleviate pain in OA patients.

Response: thank you for appreciating our study. We greatly appreciate the suggestions and advice provided.

Comment 1: Please reduce the similarity, as indicated in the attached document.

Response 1: We have further reduced the similarity. Thank you for your suggestion.

Comment 2: Clarify whether the included figures are created by the authors or sourced from other papers. If the latter, please state whether you have permission to use them or cite the original sources.

Response 2: The initial selection of images was made in good faith, with the sole purpose of providing more details on the topic at hand. To avoid any misunderstandings, we have decided to remove the images and create new ones ourselves, which we will include in the article

Comment 4: Please include at least two more therapeutic treatments or drugs, or biomaterials for a more exhaustive review.

Response 4: Thank you for your suggestion. We have highlighted the changes/additions made in the text.

Round 2

Reviewer 1 Report

Comments and Suggestions for Authors

accept

Reviewer 2 Report

Comments and Suggestions for Authors

The manuscript "The Pathogenetic Role of RANK/RANKL/OPG Signalling in Osteoarthritis and Related Targeted Therapies" has been revised according to the previous suggestions. I believe it is now ready for publication.